# Machine-Learning for Mapping and Monitoring Shallow Coral Reef Habitats

Christopher Burns [1,*], Barbara Bollard [1] and Ajit Narayanan [2]

1   Drone Lab, School of Engineering, Computer and Mathematical Sciences, Auckland University of Technology, WZ Building, Level 11, 6 St Paul Street, Auckland 1010, New Zealand; barbara.bollard@aut.ac.nz
2   School of Engineering, Computer and Mathematical Sciences, Auckland University of Technology, WZ Building, Level 11, 6 St Paul Street, Auckland 1010, New Zealand; ajit.narayanan@aut.ac.nz
*   Correspondence: cburns@aut.ac.nz

**Abstract:** Mapping and monitoring coral reef benthic composition using remotely sensed imagery provides a large-scale inference of spatial and temporal dynamics. These maps have become essential components in marine science and management, with their utility being dependent upon accuracy, scale, and repeatability. One of the primary factors that affects the utility of a coral reef benthic composition map is the choice of the machine-learning algorithm used to classify the coral reef benthic classes. Current machine-learning algorithms used to map coral reef benthic composition and detect changes over time achieve moderate to high overall accuracies yet have not demonstrated spatio-temporal generalisation. The inability to generalise limits their scalability to only those reefs where in situ reference data samples are present. This limitation is becoming more pronounced given the rapid increase in the availability of high temporal (daily) and high spatial resolution (<5 m) multispectral satellite imagery. Therefore, there is presently a need to identify algorithms capable of spatio-temporal generalisation in order to increase the scalability of coral reef benthic composition mapping and change detection. This review focuses on the most commonly used machine-learning algorithms applied to map coral reef benthic composition and detect benthic changes over time using multispectral satellite imagery. The review then introduces convolutional neural networks that have recently demonstrated an ability to spatially and temporally generalise in relation to coral reef benthic mapping; and recurrent neural networks that have demonstrated spatio-temporal generalisation in the field of land cover change detection. A clear conclusion of this review is that existing convolutional neural network and recurrent neural network frameworks hold the most potential in relation to increasing the spatio-temporal scalability of coral reef benthic composition mapping and change detection due to their ability to spatially and temporally generalise.

**Keywords:** remote sensing; machine-learning; deep-learning; coral reefs; mapping; change detection; spatio-temporal generalisation

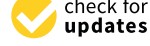



## 1. Introduction

Coral reef benthic composition maps have become an essential component in marine science and management [1]. In order for these maps to be most effective, they need to represent the underlying benthic classes as accurately as possible. The field of coral reef remote sensing science has established a number of key findings that have led to improvements in coral reef benthic composition map accuracies. These include: identifying the effects of spatial and spectral resolution [2–4]; the number of classes mapped [5–8]; the spatial and spectral similarities between benthic classes [4,8]; the use of either pixel or object-based classification methods [9]; and the in situ reference data collection method used [10]. The development of image pre-processing algorithms such as correction for absorption and scattering of light in the water column [11], sunglint removal [12], and atmospheric correction [13] have also contributed to improvements in the ability to accurately map coral reef benthic composition using remotely sensed imagery.

Another factor that affects the accuracy of a coral reef benthic composition map, which is in the control of a remote sensing scientist, is the choice of the machine-learning algorithm used to classify the coral reef benthic classes. To this end, these machine-learning algorithms use labelled samples in the form of either individual pixels or objects (grouped pixels) from within an image that are representative of specific classes in order to train an algorithm to subsequently classify all pixels or objects. In the field of coral reef benthic composition mapping, machine-learning algorithms are now more commonly used than manual delineation by expert interpretation [14], expert manual class assignment [5,15], and expert-derived ruleset development [9,16–20] This is because machine-learning algorithms are less subjective and more easily repeatable [21]. The most commonly used machine-learning algorithms for coral reef benthic composition mapping are the *k*-Nearest Neighbours (*k*-NN), Maximum Likelihood Classification (MLC), Minimum Distance to Means (MDM), Random Forest (RF), and Support Vector Machine (SVM) (Tables 1 and 2). A key advantage of these algorithms is the fact that they are able to achieve moderate to high overall accuracies with only small amounts of training data (i.e., <1000 training samples per class). To date, this has been a prerequisite for coral reef benthic mapping due to the logistical complexities inherent in acquiring in situ reference samples from a coral reef site, which are usually in the form of georeferenced benthic photographs [10].

The scarcity of coral reef benthic in situ reference data is perhaps the reason almost all related publications, to date, use training and testing data derived from the same specific reef, or reef area, within the extent that they are mapping. While this is suitable for mapping coral reef benthic composition at the target locations, where in situ reference data are present, the inherent bias between the training and testing samples will result in the machine-learning algorithm overfitting on this localised data. Therefore, it is unlikely that the trained algorithm can then generalise to mapping the coral reef benthic composition of a new reef (spatial generalisation), or even new imagery of the same reef (temporal generalisation) since remote sensing conditions may vary between images. The inability to spatially or temporally generalise therefore affects the spatio-temporal scalability of coral reef benthic mapping by limiting the target extents to only those reefs or reef areas that contain in-situ reference data.

In addition to mapping coral reef benthic composition at one point in time, ongoing monitoring is essential for conservation management. Coral reef benthic composition monitoring using remotely sensed imagery, to date, has relied primarily on post-classification comparison change detection (PCCCD), which classifies each individual image separately [14,22–24] before subsequently overlaying the classified images in order to identify where change has occurred. The accuracy of PCCCD methods is therefore dependent upon the accuracy of each individual classified map in the sequence. Since each map in the multi-temporal sequence is usually classified using similar machine-learning algorithms as those used in coral reef benthic composition mapping, the same primary limitation in terms of spatio-temporal scalability, the inability to generalise, is also present in PCCCD methods used for coral reef benthic change detection. The inability to generalise is becoming more pronounced given the rapid increase in the availability of high temporal (daily) and high spatial resolution (<5 m) multispectral satellite imagery [25].

Coincident with the increase in the availability of multispectral satellite imagery are advancements in deep learning frameworks [26,27]. Of particular interest to remote sensing scientists are convolutional neural networks (CNNs), which, in the field of land cover mapping, have demonstrated superior overall accuracies compared to traditional machine-learning classification algorithms such as object-based *k*-nearest neighbour (OBIA–KNN), object-based support vector machine (OBIA–SVM), and object-based random forest (OBIA–RF) [28,29], as well as spatio-temporal generalisation [30]. CNNs have only recently been used in the field of coral reef benthic mapping [31–33] but have already demonstrated high overall accuracies and spatio-temporal generalisation [32]. In relation to change detection,

Long Short-Term Memory networks (LSTMs) and recurrent convolutional neural networks (ReCNN) have been applied to binary (changed or unchanged) and multi-class land cover change detection and achieved superior overall accuracies compared to traditional land cover change detection methods, and they have demonstrated binary spatio-temporal generalisation [34,35]. To the best of our knowledge, however, LSTMs have not yet been applied to coral reef benthic change detection.

This review focuses on the most commonly used machine-learning algorithms applied to map coral reef benthic composition and detect changes over time using multispectral satellite imagery in order to identify their advantages as well as the primary limitations impacting their spatio-temporal scalability. Recent publications that have used CNNs for coral reef benthic mapping, and LSTMs in the field of land cover change detection are subsequently reviewed with a focus on their potential utility for increasing the spatio-temporal scalability of coral reef benthic mapping. Three keywords, 'coral reef habitat mapping', 'coral reef mapping using machine learning', and 'coral reef change detection' were searched in ScienceDirect and Google Scholar. For each keyword the first 100 results were screened in ScienceDirect and the first 10 pages in Google Scholar. No date range for publications was applied. We limited publications to only those that used multispectral satellite imagery and mapped shallow coral reef habitat classes. Multispectral satellite imagery was chosen since it is the most easily accessible imagery type for researchers, covers the largest area globally, and has the highest temporal frequency compared to airborne and/or UAV imagery. Publications focusing only on seagrass or coral reef geomorphological classes were excluded. Book chapters, theses, reports, and reviews were also excluded. From 600 screened results, 87 publications were deemed relevant based on the title and/or the abstract. Of these, 6 could not be accessed and 15 after being reviewed did not include enough detail to be included, as seen in Tables 1–3, leaving a total of 66 publications. A further five additional publications were manually searched since they were referenced a number of times within these 66 publications and therefore deemed important to include in this review. These publications were: [7,9,15,17,21]. One additional publication [33], was brought to our attention because it used a CNN for coral reef benthic mapping. In total, 72 publications based on coral reef mapping and change detection have been included in this review.

For CNN publications used in Sections 2.2 and 2.2.1, in addition to recurrent neural networks (RNNs) in Section 2.3.2, a keyword search methodology was not applied. CNNs were included in this review based on the fact they are starting to be used for coral reef benthic mapping. RNNs were included because they have demonstrated spatio-temporal generalisation in the field of land cover change detection, which current coral reef change detection methods have not demonstrated.

## 2. Machine-Learning Algorithms Applied to Coral Reef Benthic Mapping Using Multispectral Satellite Imagery

### 2.1. Pixel-Based Machine-Learning Classification Algorithms

Pixel-based machine-learning classification algorithms applied to mapping coral reef benthic composition, classify each individual pixel within an image as being one of a certain number of classes (i.e., coral, algae, and sand). In order for pixel-based algorithms to be feasible, two assumptions need to be met. First, each individual pixel needs to be represented by only one benthic class, meaning the spatial resolution of the pixel is higher than or similar to the target object. Secondly, pixels representing each class need to have similar spectral reflectance values to each other (i.e., all coral pixels need to have similar spectral reflectance signatures to each other, which are different to the spectral reflectance signatures of sand and other classes) [36]. For coral reef benthic composition mapping, the main pixel-based machine-learning algorithm used is MLC, based on the fact it is used by 47% of publications in Table 1 (In relation to calculating the number of publications that use each specific machine learning algorithm in Tables 1–3, only one algorithm per publication is used. When a publication compares multiple different machine learning algorithms, only

the algorithm that achieves the highest overall accuracy is used in the calculation. This was done in order to ensure that publications that use one machine learning algorithm and those that are comparing multiple different ones are weighted equally in the calculation). MLC, which first assumes the data are normally distributed, takes the mean vector and covariance of each classes' spectral values into account before subsequently determining the membership of individual pixels to each class based on the highest statistical probability (maximum likelihood). Figure 1 illustrates equiprobability contours forming probability density regions around the mean of the class training samples that are used in order to determine the probability of pixel *x* belonging to each class. In this example pixel *x* would be classified as belonging to the coral class.

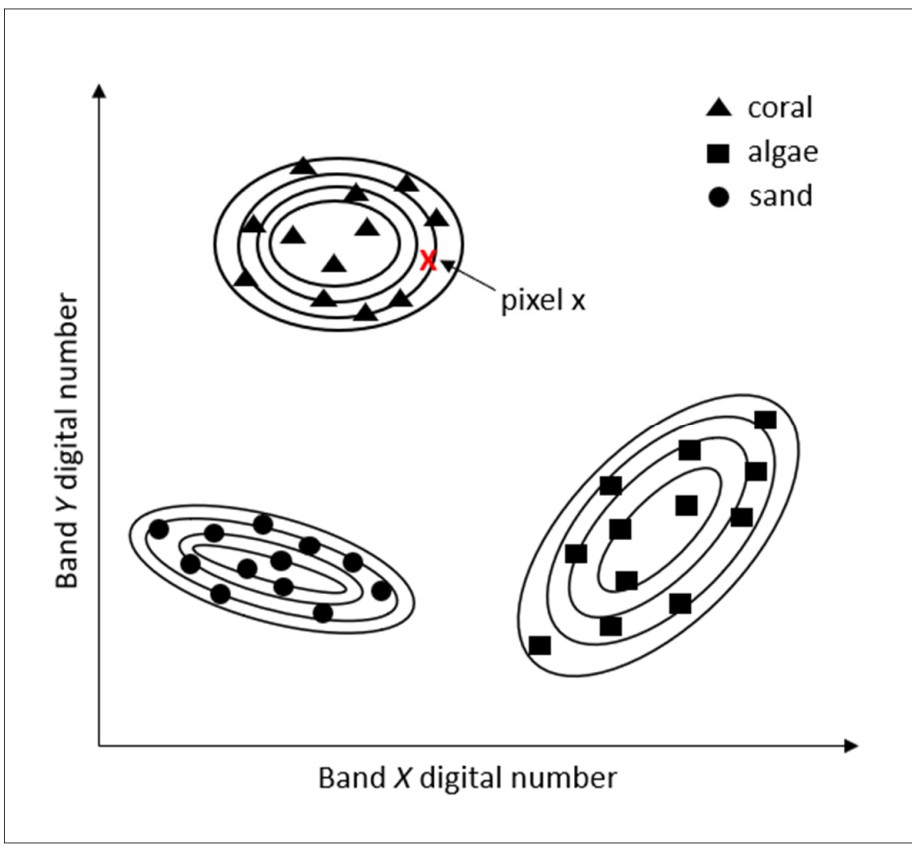

**Figure 1.** Basic conceptualisation of maximum likelihood classification illustrating 3 different classes with equiprobability contours.

Moderate to high overall accuracies can be achieved using pixel-based MLC to map coral reef benthic classes (Table 1) (All overall accuracies reported in this review, except for those in Section 2.3.2 have been rounded to the nearest whole number since there was not a consistent level of precision between all publications (i.e., some recorded overall accuracy to the nearest one decimal place while others to the nearest whole number)). Compared to the second most commonly used machine-learning algorithms in Table 1, which are RF, SVM, and MDM, each being used by 12% of publications, MLC has demonstrated higher overall accuracies when directly compared to MDM in Hossain et al. [37]. Wicaksono and Aryaguna [8], and Chegoonian et al. [38], however, found that RF and SVM, respectively, achieved higher overall accuracies when directly compared to MLC (Table 1).

**Table 1.** Pixel-based coral reef benthic mapping publications.

| Authors | Sensor/Spatial Resolution | Pixel or Object-Based | Classification Algorithm(s) | Supervised or Unsupervised | Number of Benthic Habitat Classes | Accuracy (Overall Accuracy) |
|---|---|---|---|---|---|---|
| [39] | WorldView-3 (1.2 m). | Pixel-based. | SVM. | Supervised. | 15 | 79% |
| [37] | QuickBird (2.4 m). | Pixel-based. | MLC, MD, *k*-NN, Parallelepiped classification (PP), and Fisher (F); then ensemble. Classification using Majority Voting (MV), Simple Averaging (SA), and Mode Combination (MC). | Supervised. | 4 | 55% (MLC), 53% (MD), 54% (KNN), 41% (PP), 47% (F), 83% (MV), 71% (SA), and 68% (MC). |
| [8] | WorldView-2 (1.9 m). | Pixel-based. | MLC, RF. | Supervised. | 2–26 | 74.01–22.15% (mean MLC), 95.97–76.83% (mean RF). |
| [40] | PlanetScope (3.7 m). | Pixel-based. | RF. | Supervised. | 4 | 78% (Cemara Islands based on 500 trees), 61% (Gelang Island based on 500 trees); 79% (Cemara Islands based using Log and Entropy function), 61% (Geland Island using Square Root and Gini function). |
| [41] | Landsat-8 OLI (30 m). | Pixel-based. | Linear Discriminant Analysis (LDA). | Supervised. | 4 | 80% (Palmyara Atoll), 79% (Kingman Reef), 69% (Howland Island), 71% (Baker Island Atoll), and 74% (Combined). |
| [42] | Planet Dove (4.7 m). | Pixel-based. | ISODATA classification. | Unsupervised. | 8 | 63% |
| [38] | Landsat-8 (30 m). | Pixel-based. | MLC, SVM, ANN. | Supervised. | 4 | Lizard Island; 72% (ANN), 67% SVM, 67% MLC; Qeshm and Larak Islands; 58% (ANN), 68% (SVM), 66% (MLC). |
| [43] | IKONOS (4 m). | Pixel-based. | MLC. | Supervised. | 6 | 82% |
| [10] | QuickBird-2 (2.4 m) local. | Pixel-based. | MDM. | Supervised. | 21 | Suva site: 69% (photo transect), 65% (spot check). |
| [44] | IKONOS (4 m). | Pixel-based. | MLC. | Supervised. | 9 | 89% (Bawe) and 80% (Chumbe). |
| [45] | QuickBird (2.4 m). | Pixel-based. | MLC. | Supervised. | 6 | 67% (no water column correction), 89% (with water column correction. |

**Table 1.** *Cont.*

| Authors | Sensor/Spatial Resolution | Pixel or Object-Based | Classification Algorithm(s) | Supervised or Unsupervised | Number of Benthic Habitat Classes | Accuracy (Overall Accuracy) |
|---------|---------------------------|-----------------------|-----------------------------|----------------------------|-----------------------------------|------------------------------|
| [46] | Landsat-7 ETM+ (30 m). | Pixel-based. | Ensemble of hybrid SVM Classifiers. | Supervised. | 5 | 89% |
| [47] | QuickBird-2 (2.4 m), Landsat 5 TM (30 m). | Pixel-based. | MDM. | Supervised. | 10–21 | 25–62% |
| [48] | Landsat 5 TM (30 m). | Pixel-based. | MLC. | Supervised. | 7 | 76% |
| [7] | IKONOS (4 m). | Pixel-based. | MLC. | Supervised. | 4, 8, 13 | 75% (4 classes), ~65% (8 classes), and 50% (13 classes). |
| [49] | Landsat TM (30 m). | Pixel-based. | MLC followed by contextual editing. | Supervised. | 4, 8, and 13 | ~60%, ~40%, and ~25%, respectively. |
| [4] | Landsat TM (30 m). | Pixel-based. | MLC. | Supervised. | 4, 6, 9 | ~57% (4 classes), ~53% (6 classes), and ~50% (9 classes). |

For both Tables 1 and 2, in publications that compared multiple different sensors, only the sensors with the highest overall are in this table, since this is the most common metric reported in the majority of these publications. Furthermore, we aimed to make overall accuracies as consistent as possible between publications, which was difficult since many compared OA of different sensors, spatial resolution, classification algorithms etc., therefore, as a general rule we aimed to report the highest OA from each publication. This resulted in some including more information than others, which is a shortcoming of this review. Only the overall accuracies from the IKONOS sensor used in Mumby and Edwards. Ref. [7] are included since it was thought to be the most relevant to that particular publication even though it was not the highest overall accuracy achieved. While Andréfouët et al. [5] mapped multiple sites, the overall accuracies reported in this table are derived from their abstract, which is thought to be the average overall accuracy reported based on the number of classes mapped. When more than two accuracy metrics were used, only the overall accuracies were included.

In relation to spatio-temporal generalisation, only one publication in Table 1 demonstrated spatial generalisation [41] and none demonstrated temporal generalisation. Gapper et al. [41] used a pixel-based Linear Discriminant Analysis (LDA) classification algorithm applied to mapping two coral reef benthic habitat classes (coral and algae/sand) and two background classes (land/cloud and deep water) at four different coral reef sites using Landsat-8 multispectral imagery with a spatial resolution of 30 m. Their results showed the pixel-based LDA achieved an overall accuracy of 80% when trained and tested on the same reef, and 79%, 69%, and 71% when tested on three different reefs that the LDA had not been trained on; therefore, demonstrating spatial generalisation.

Although moderate to high overall accuracies can be achieved using pixel-based machine-learning algorithms, there are three inherent problems that affect pixel-based classification. The first is the problem of a mixed pixel, whereby a single pixel may contain multiple benthic classes, as illustrated in Figure 2a. The mixed pixel problem is a serious challenge for mapping coral reef benthic composition because of the spatial heterogeneity inherent in coral reef benthic habitats. The second problem is known as the 'salt and pepper effect' whereby single pixels are classified differently to neighbouring pixels surrounding them, since no information from the neighbouring pixels is considered during per-pixel classification [50]. The third problem is pixel redundancy resulting from the spatial resolution being much finer than the target objects as shown in Figure 2c, where a large number of pixels represent the same target feature [51,52]. Pixel redundancy is becoming more pronounced given the increasingly higher spatial resolutions of satellite imagery (<5 m).

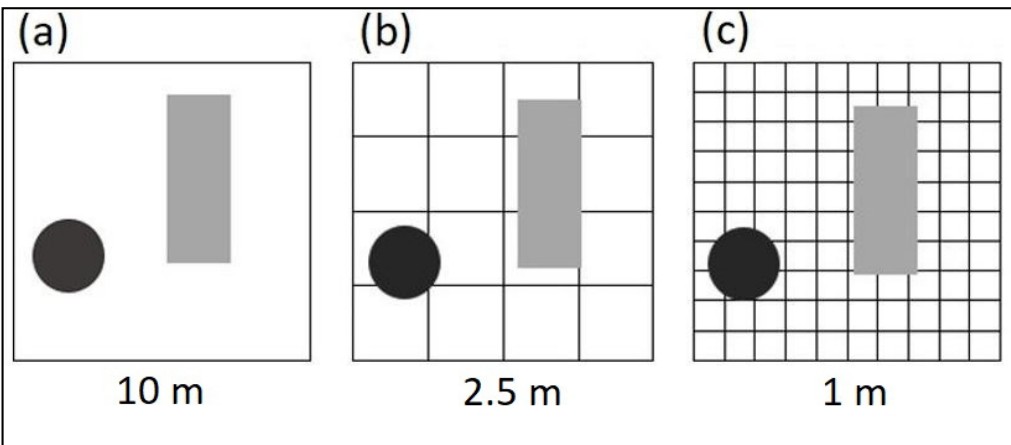

**Figure 2.** Illustration depicting the effect of spatial resolution on target objects. (**a**) At 10 m pixel resolution it is evident that pixel resolution is much lower (meaning the pixel is larger) than the objects resulting in a mixed class pixel. (**b**) At 2.5 m resolution the objects are closer to actual pixel resolution and are clearly separated meaning this 2.5 m resolution would not result in the mixed pixel problem and pixel redundancy would not be an issue. (**c**) At 1 m resolution objects are larger than pixels resulting in pixel redundancy. Illustration is based on Figure 1 in Blaschke [51].

Object-Based Image Analysis

An alternative approach to pixel-based classification is Object-based Image Analysis (OBIA), which is a segmentation approach that is not as severely affected by mixed-pixels, salt and pepper effects, and pixel redundancy. OBIA has also been referred to as Geographic Object Based Image Analysis (GEOBIA) [53] and Object-oriented Image Analysis [54]. The foundation for OBIA is image segmentation that dates back to the 1970s [55,56]. OBIA considers contextual information, which, in addition to spectral information, includes spatial dimensions of shape and compactness in order to generate relatively homogenous segments of grouped pixels that are semantically significant [51] (Figure 3b,c). Compared to individual pixels, these segmented objects have the advantage of containing much more spectral information such as the mean, median, minimum, and maximum spectral values per band in addition to mean ratios and variance [51].

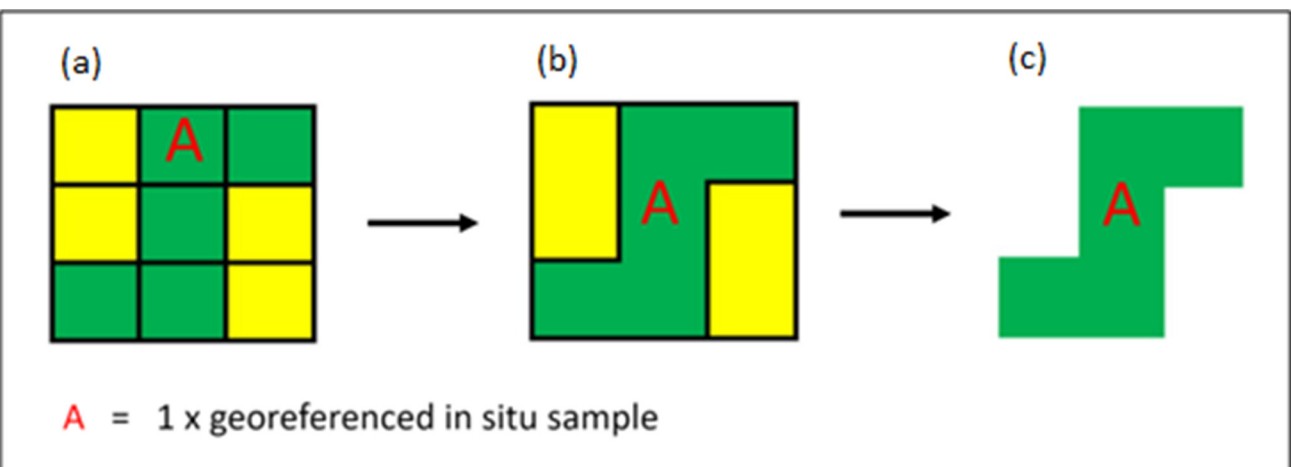

**Figure 3.** Example of object-based segmentation applied to an illustration depicting pixels. (**a**) Consists of 5 × green pixels and 4 × yellow pixels, (**b**) shows the result of object-based segmentation where all 5 × green pixels have been grouped into 1 × green object and 2 × yellow objects. The red A represents a georeferenced in situ reference sample which in (**a**) represents 1 × pixel, however, after OBIA segmentation in (**b**,**c**) it now represents 1 × green object consisting of 5 × grouped pixels.

Segmenting individual pixels into objects can be done using specialised software such as Trimble eCognition or ENVI [6,9,15–18,20,21,57–64]. Other approaches to generating segmented objects include unsupervised ISODATA clustering [65,66], unsupervised *k*-means cluster analysis [67], bag of features [68], seed pixel regional growing technique [69], texture analysis [70], image patches [33], and manual polygon digitisation [71,72].

It is important to note that OBIA using specialised software such as Trimble eCognition requires a user to determine a scale parameter. This scale parameter determines the output object size, which is difficult to identify since semantically significant regions are found at different scales [73]. While there have been techniques developed to objectively identify the optimal scale parameter, such as the estimation of scale parameter tool (ESP) developed by Dragut et al. [74], most coral reef benthic mapping publications that use OBIA determine the scale parameter based on subjective trial and error [6,21,59].

In the field of coral reef benthic composition mapping, OBIA has been applied on local [6,9,16,20,21,57,60–64,71] and regional scales [15,17,18,59]. In order to classify objects derived from OBIA segmentation, expert class assignment, whereby the map producer assigns classes to objects manually, has been shown to be an effective method for mapping coral reef benthic habitat classes on regional scales [15]. The limitations associated with expert class assignment, however, are the subjective nature of the map producer's classification ability, the limitation in the number of objects a map producer can manually label, and the difficulty in replicating this same classification with a new map producer. These issues therefore limit the spatio-temporal scalability of this approach. Phinn et al. [9] demonstrated how OBIA followed by expert-driven membership rulesets, which are used to assign classes to objects, can achieve overall accuracies of up to 78% for mapping 13 benthic habitat classes using Quickbird-2 multispectral satellite imagery with a spatial resolution of 2.4 m. OBIA followed by expert-driven membership rulesets is the most commonly used approach in Table 2, which is used by 23% of publications. Using expert-driven membership rulesets for classification, however, is also relatively subjective. For example, Phinn et al. [9] developed a total of 36 membership rules (based on variables such as brightness, standard deviation, blue/green band ratio, and others) with each rule containing an individual threshold that is iteratively determined by comparing the resultant segmented objects with expert knowledge of the reef, image interpretation, and references to in situ reference data in order to label reef scale classes (i.e., land, deep water, and shallow reef), geomorphic scale classes (i.e., outer reef flat and shallow lagoon), and benthic community scale classes (i.e., algae, seagrass, sand, and rock) [9].

Apart from expert class assignment or developing expert driven rulesets in the field of coral reef benthic composition mapping, one of the most commonly used machine-learning algorithms applied to classifying segmented objects derived from OBIA is the *k*-NN algorithm that is used by 12% of the publications in Table 2. The *k*-NN algorithm classifies segmented objects based on the class most represented by their *k* nearest neighbours [75,76]. *K* is a user-defined parameter that is the number of nearest neighbouring objects that are included in the majority voting process. When *k* is equal to one it is referred to as the Nearest Neighbour (NN) [21] and simply classifies an object based on the class of its nearest neighbour. The choice of *k* affects accuracy, as illustrated in Figure 4, whereby a value of *k* = 1 would result in *x* being classified as algae while a value of *k* = 5 would classify *x* as coral.

The accuracy of the *k*-NN algorithm is also affected (when not detected and separated) by the presence of class outliers [77]. Class outliers will be defined here as objects with classes other than their own surrounding them. These outliers can be the result of an insufficient number of training samples associated with the outlier class or skewed class distributions. Datasets used to train machine-learning algorithms for classifying coral reef benthic composition may be prone to containing class outliers since coral reef benthic classes are typically heterogenous in spatial distribution. Further, the logistical complexities of acquiring in situ reference samples from a coral reef location limits the quantity of samples that can be acquired.

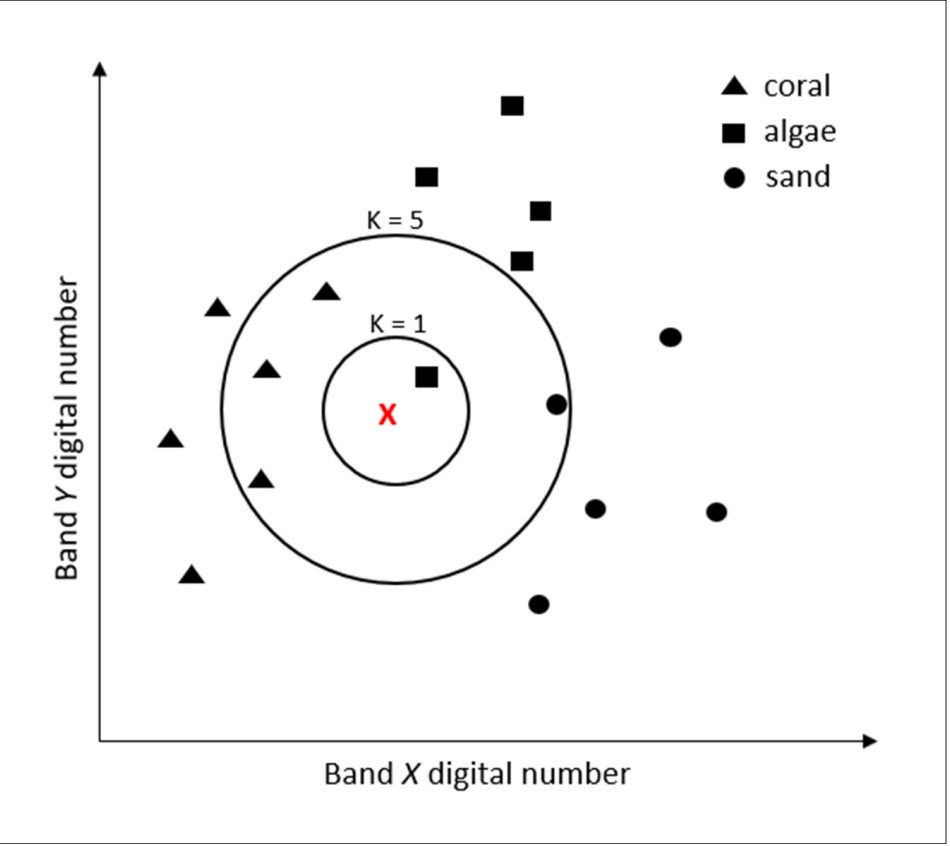

**Figure 4.** Basic conceptualisation of *k* nearest neighbour classification illustrating 3 classes. Circles show the class when *k* = 1 and when *k* = 5.

To address the fact that coral reef benthic habitat complexity varies across different coral reef geomorphologic zones [5], hierarchical OBIA segmentation approaches were developed by [9,15,21], which first segments a reef into different geomorphic zones and then re-segments each of these geomorphic zones individually in order to generate segments representing benthic habitat classes within each. Roelfsema et al. [21] compared a *k*-NN classification algorithm to the expert driven membership rulesets classification approach of Phinn et al. [9], applied to the same reef using the same in situ reference data and Quickbird-2 multispectral imagery. The results showed the object-based *k*-NN achieved an overall accuracy of 62% based on seven benthic classes, while the membership rulesets classification approach achieved an overall accuracy of 78% based on eleven benthic classes (Table 2). Although the overall accuracy of the object-based *k*-NN is lower, in addition to being less subjective, Roelfsema et al. [21] estimated that their hierarchical object-based *k*-NN approach is around twenty times faster to develop compared to Phinn et al. [9].

Another of the most commonly used object-based machine-learning classification algorithms applied to mapping coral reef benthic composition is the SVM algorithm, which is used in 17% of publications in Table 2. The modern formulation of an SVM, developed by Vapnik and Cortes [78], classify inputs by identifying decision boundaries that split input data points into two spaces that correlate to two different classes (Figure 5). This is done by mapping the input data to a new high-dimensional representation where the decision boundary (expressed as a hyperplane) is computed by maximising the distance between the decision boundary and the nearest data points from each class [79]. In order to find good decision boundaries, rather than explicitly computing the coordinates of the points in the new representation space, which is often computationally intractable, a kernel function is used. This maps two points in the initial representation space to the distance between these points in the target representation space, which is more efficient [79]

(Figure 5). Object-based SVM have been proven to achieve higher overall accuracies when directly compared to *k*-NN [63,68,72] (Table 2).

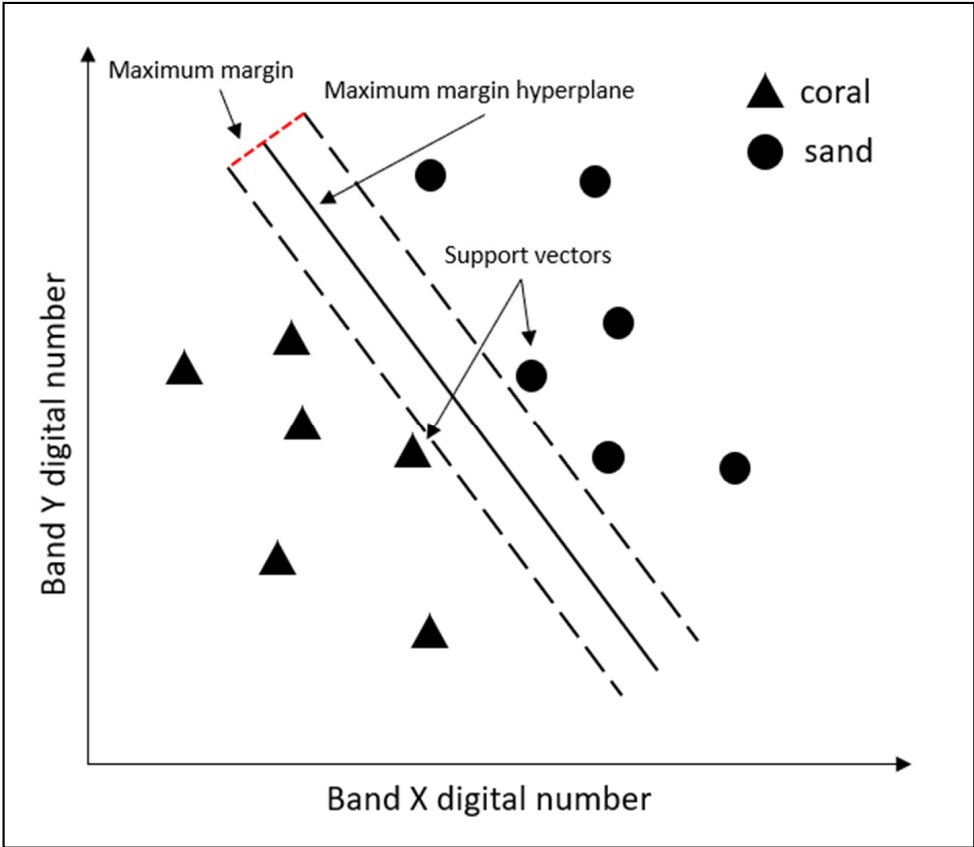

**Figure 5.** General conceptualisation of an SVM.

**Table 2.** Object-based coral reef benthic mapping publications.

| Authors | Sensor/Spatial Resolution | Pixel or Object-Based | Classification Algorithm(s) | Supervised or Unsupervised | Number of Benthic Habitat Classes | Accuracy (Overall Accuracy) |
|---|---|---|---|---|---|---|
| [58] | Sentinel-2 (10 m). | Object-based. | OBIA–RF followed by expert-driven membership rulesets. | Supervised. | 4 | 62% |
| [70] | Sentinel-2 (10 m). | Object-based. | Mean Texture Analysis followed by either RF or SVM. | Supervised. | 4 | 71% (RF, highest), 73 (SVM, highest). |
| [59] | Worldview-2 (1.9 m), Planet Dove (5 m), Sentinel-2 (10 m), Landsat-8 (15 m). | Pixel-based and object-based. | Pixel-based RF and OBIA–RF followed by expert-driven membership rulesets. | Supervised. | 8 | 78% (mean). |

**Table 2.** *Cont.*

| Authors | Sensor/Spatial Resolution | Pixel or Object-Based | Classification Algorithm(s) | Supervised or Unsupervised | Number of Benthic Habitat Classes | Accuracy (Overall Accuracy) |
|---|---|---|---|---|---|---|
| [17] | Landsat OLI (15 m). | Object-based. | OBIA followed by expert-driven membership rulesets. | Supervised. | 6 | 50% |
| [31] | WorldView-2 (1.9 m), Sentinel-2 (10 m). | Object-based. | LAPDANN. | Supervised. | 10 | 86% (trained/tested on same reef), 47% (trained/tested on data from Indian Ocean and Pacific Ocean simultaneously). |
| [32] | WorldView-2 (1.9 m), PlanetScope (3.7 m). | Object-based and pixel-based. | FCN–KNN, VGG16–FCN, DeepLab, SharpMask. | Supervised. | 9 | WorldView-2 imagery: 84% (FCN–KNN), 80% (VGG16–FCN), 81% (DeepLab), 80% (SharpMask); PlanetScope imagery: 73% (FCN–KNN), 73% (VGG16–FCN), 73% (DeepLab), 71% (SharpMask); Generalisation tests: 85% (FCN–KNN), 83% (VGG16–FCN), 78% (DeepLab), 82% (SharpMask). |
| [80] | WorldView-2 (1.9 m), Gaofen-2 (3.2 m). | Object-based. | CNN–SVM, CNN–RF, CNN, RF, SVM. | Supervised. | 4 | WorldView-2 data set 1: 92% (CNN–SVM), 91% (CNN–RF), 91% (CNN), 90% (RF), 89% (SVM); WorldView-2 data set 2: 86% (CNN_SVM), 85% (CNN–RF), 85% (CNN), 82% (RF), 84% (SVM) Gaofen-2 data set: 91% (CNN–SVM), 88% (CNN–RF), 89% (CNN), 87% (RF), 88% (SVM). |
| [33] | QuickBird (0.6 m) (benthic), GeoEye-1 (0.5 m) (seagrass). | Object-based. | CNN. | Supervised. | 7 benthic, 4 seagrass. | 90% (benthic), 91% (seagrass). |
| [67] | Sentinel-2 (10 m). | Object-based. | MD followed by post-classification filtering. | Supervised. | 17 (incl. 5 non-coral reef benthic classes (i.e., man-groves, beach). | 77% |

**Table 2.** *Cont.*

| Authors | Sensor/Spatial Resolution | Pixel or Object-Based | Classification Algorithm(s) | Supervised or Unsupervised | Number of Benthic Habitat Classes | Accuracy (Overall Accuracy) |
|---|---|---|---|---|---|---|
| [72] | WorldView-2 (1.9 m). | Object-based. | MLC, Neural Network (NN), SVM. | Supervised. | 5 | 86% (MLC), 87%(NN), 93% (SVM). |
| [15] | WorldView-2 (1.9–2.4 m). | Object-based. | OBIA followed by manual class assignment. | Expert-derived. | Atlantic sites: 7 (aggregated Benthic cover type and geo-morphology classes (i.e., Fore Reef Sediment with Algae), Non-Atlantic sites: 16. | 81% (Atlantic sites), 90% (non-Atlantic sites). |
| [60] | WorldView-2 (1.9 m). | Object-based. | OBIA–RF, OBIA–Classification Tree Analysis (OBIA–CTA), OBIA–SVM. | Supervised. | 14 | 89% (RF), 78% (CTA), 76% (SVM). |
| [61] | Planet Dove (3 m). | Object-based. | OBIA–KNN. | Supervised. | 11 | 82% |
| [57] | GeoEye-1 (2 m). | Object-based. | OBIA and Jeffries–Matusita distance measure. | Supervised. | 175 | 72% |
| [21] | QuickBird-2 (2.4 m). | Object-based. | OBIA–KNN. | Supervised. | 7 | 62% |
| [68] | QuickBird (2.4 m). | Object-based. | Bag of Features (BOF) followed by either Bagging (BAG), KNN, or SVM then lastly a Weighted Majority Voting (WMV). | Supervised. | 4 | 80% (BAG), 81% (KNN), 86% (SVM), 89% (WMV). |
| [71] | WorldView-2 (1.9 m). | Object-based. | SVM. | Supervised. | 5 | 78% |
| [16] | Sentinel-2 (10 m). | Object-based. | OBIA with expert-driven membership rulesets. | Supervised. | 6 | 49% |
| [19] | Landsat 8 (15 m). | Object-based. | OBIA with expert-driven membership rulesets. | Supervised. | 5 | 33% |

**Table 2.** *Cont.*

| Authors | Sensor/Spatial Resolution | Pixel or Object-Based | Classification Algorithm(s) | Supervised or Unsupervised | Number of Benthic Habitat Classes | Accuracy (Overall Accuracy) |
|---|---|---|---|---|---|---|
| [69] | Landsat 7 ETM+ (30 m), Landsat 8 (30 m). | Object-based. | Seed pixel regional growing. | Supervised. | 3 coral reef benthic and 2 non-benthic (i.e., land and human habitats). | 75–99.7% based on 10 sites. |
| [20] | WorldView-2 (1.9 m). | Object-based. | OBIA with expert-driven membership rulesets. | Supervised. | 4 | 76% |
| [62] | WorldView-2 (1.9 m). | Object-based. | OBIA-multinomial logistic discrete choice models. | Supervised. | 8 benthic and 3 non-benthic (i.e., terrestrial vegetation). | 85% (Vanua Vatu site). |
| [63] | Landsat 8 OLI (30 m). | Object-based. | OBIA–SVM, OBIA–RT, OBIA–DT, OBIA–KNN, OBIA–Bayesian. | Supervised. | 7 | 73% (OBIA–SVM), 68% (OBIA–RT), 67% (OBIA–KNN), 66% (OBIA–Bayesian), and 56% (OBIA–DT). |
| [18] | QuickBird-2 (2.4 m), IKONOS (4 m). | Object-based. | OBIA with expert-driven membership rulesets. | Supervised. | 14–17 (individual reefs), 20–30 (reef systems). | 52–75%. |
| [9] | QuickBird-2 (2.4 m). | Pixel-based and object based. | OBIA with expert-driven membership rulesets; pixel-based MDM. | Supervised. | Heron Reef: 13 Ngderack Reef: 11 Navakavu Reef: 17. | 78% (Heron Reef, object-based), 52% (Ngderack Reef, object-based), 65%, 57% (Navakavu Reef, object-based and pixel-based, respectively). |
| [64] | QuickBird-2 (2.4 m), IKONOS (4 m). | Object-based. | OBIA with expert-driven membership rulesets. | Supervised | 22 benthic and 3 non-benthic (i.e., cloud). | 67%. |
| [6] | QuickBird (0.6 m Pan-sharpened). | Pixel-based and object-based. | Pixel-based MLC and contextual editing; OBIA–NN. | Supervised. | 5, 7, and 11. | 59–77% (MLC), 61–76% (contextual editing), and 81–90% (OBIA–NN). |
| [66] | Landsat TM (30 m). | Object-based. | Unsupervised ISODATA Classification. | Unsupervised. | 7 | 74% |
| [5] | IKONOS (4 m), Landsat 7 ETM+ (30 m). | Object-based (unsupervised segments and ground-truthed polygons). | Unsupervised segmentation followed by expert class assignment (applied to 2 reefs); MLC (applied to 7 reefs). | Unsupervised (2 reefs) and supervised (7 reefs). | 4–5, 7–8, 9–11, >13. | 77% (4–5 classes), 71% (7–8 classes), 65% (9–11 classes), and 53% (> 13 classes). |

**Table 2.** *Cont*.

| Authors | Sensor/Spatial Resolution | Pixel or Object-Based | Classification Algorithm(s) | Supervised or Unsupervised | Number of Benthic Habitat Classes | Accuracy (Overall Accuracy) |
|---|---|---|---|---|---|---|
| [65] | IKONOS (4 m). | Object-based. | MLC. | Supervised. | 5 | 90% (Half Moon Bay), 89% (Tabyana Bay). |

Unsupervised ISODATA clustering [65,66], unsupervised K-means cluster analysis [67], bag of features [68], seed pixel regional growing technique [69], image patches [33], manual polygon digitisation [71], and digitising regions of interest [57], have been grouped into the 'object-based' even though these polygons are not derived by utilizing a segmentation algorithm such as those derived from OBIA (eCognition for example). Additionally, the multi-scale feature extraction approach using bidimensional empirical mode decomposition (BEMD) [80] has also been grouped into 'object-based.' Publications that included both pixel-based and object-based classifications are included in Table 2 only, rather than Table 1 as well.

Second to the object-based SVM, and equal with the *k*-NN algorithm in terms of being used in Table 2, is the RF algorithm that is used by 12% of publications. RFs construct a wide variety of uncorrelated decision trees during training, with each individual tree consisting of a random sample (bootstrapped) of the training data, and then takes the aggregate (mode) of these individual trees as the output classification [81,82] (Figure 6). RFs are less prone to overfitting compared to individual decision trees [82] and they use the 'stochastic discrimination' method [83], which has been proven to reduce overfitting and a lack of generalisation properties [84–86]. These properties lead to RF algorithms gaining accuracy as they become bigger and more complex [81,83].

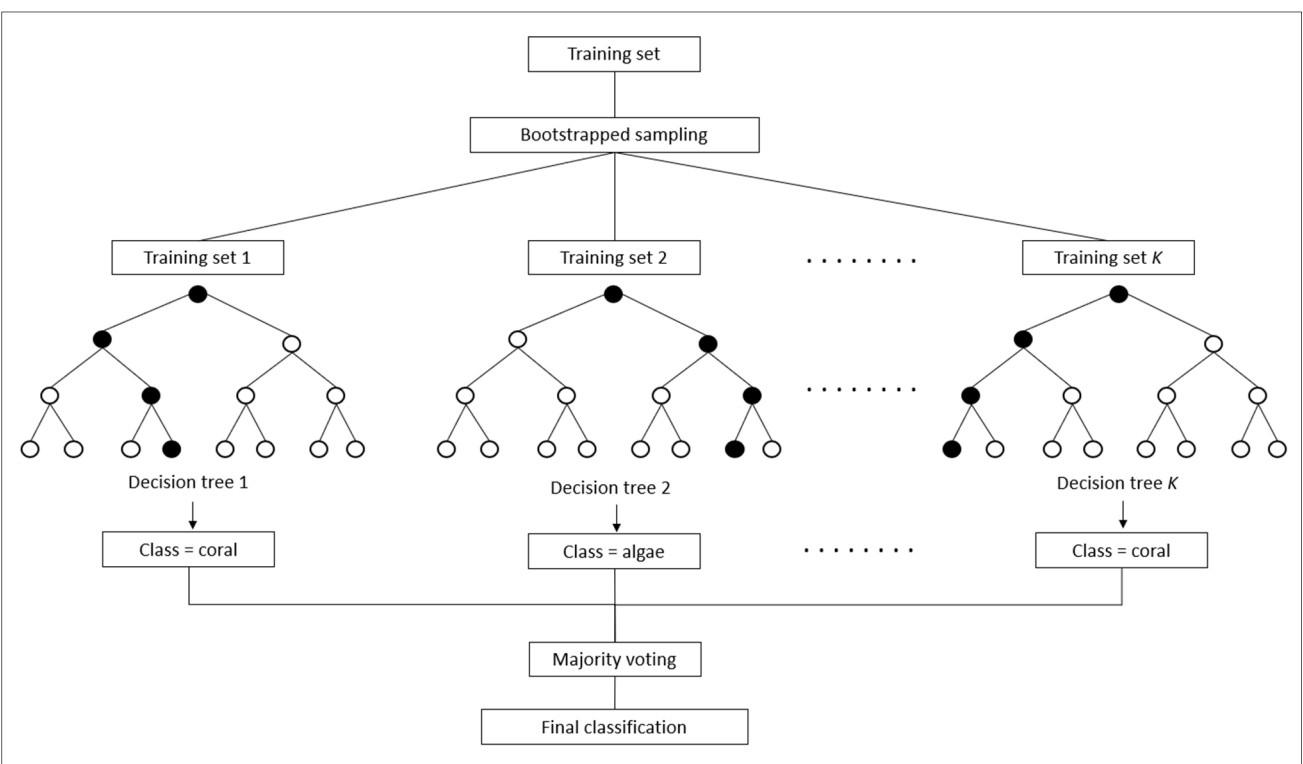

**Figure 6.** Basic conceptualization of random forest classification.

In Table 2 there are two publications that have directly compared object-based RF to object-based SVM algorithms. One found the object-based RF algorithms to be inferior in terms of overall accuracy compared to an object-based SVM [70], while the other found the object-based RF to be superior [40] (Table 2). In relation to spatio-temporal generalisation, Lyons et al. [59], as part of the Allen Coral Atlas project that aims to map shallow

water coral reefs on a global scale, adopted a similar hierarchical mapping approach to Roelfsema et al. [21] in order to map fourteen geomorphic classes and eight benthic habitat classes on individual (~200 km² of reef) to regional scales (~200,000 km² of reef). A key difference, however, is that Lyons et al. [59] used an object-based RF algorithm rather than a *k*-NN algorithm, which is used by Roelfsema et al. [21]. When applied to mapping benthic habitat classes at different scales (individual reefs and regional scales), their frameworks mean that the overall accuracy is 78%. This overall accuracy, however, is based on validation data derived from within the regions where in situ training and validation data samples are present. Although this overall accuracy has been inferred to areas that contain no in situ training or validation data, for example, when scaling up from individual reefs to regional scales such as the entire GBR region, the accuracy for such areas is uncertain. Therefore, while their framework can be transferred to new areas provided that coincident in situ reference data (or expert derived polygons) is available, it is yet to be determined if it can spatially or temporally generalise to areas where no in situ reference training or validation data are present.

### 2.2. Convolutional Neural Networks

In relation to land cover mapping using remotely sensed imagery, convolutional neural networks (CNNs) have emerged as new algorithmic approaches for object detection [87], image classification [88], and image segmentation [89]. CNNs are a class of deep learning algorithms, which, put generally, take an input image (Figure 7) and convolve it through a series of successive layers in order to learn a hierarchical feature set that can subsequently be used for supervised classification at the final layer of the framework (Figure 8). In relation to remotely sensed imagery, the input layer is a tensor with shape dimensions equal to the image height multiplied by image width, and the number of channels (spectral bands) For instance, Figure 7 illustrates an example input from a multispectral satellite image with dimensions of 256 pixels (image height) × 256 pixels (width) × 4 channels (red, green, blue, and near-infrared bands). The input sample starts as 262,144 individual pixels, each consisting of four spectral bands. If there are 65,536 pixels representing 'coral' (white), this then leaves 196,608 pixels that are 'non-coral' (black), which are assigned a value of 0. This particular input sample will therefore consist of 65,536 individual pixels each with four spectral values. This input sample differs from the pixel-based samples in Figure 2, which each only contain one pixel with four spectral values, and also from the OBIA sample in Figure 3, since it does not contain grouped information (i.e., mean, minimum, maximum spectral values etc.) for the object itself, although CNNs do still extract contextual information from the input sample.

The fundamental building blocks of a CNN are the convolution operations that work by applying multiple filters each consisting of weights that either learn to find local patterns such as edges and textures or have been predefined to do so. Each filter is usually 3 × 3 in size and is applied across the width, height, and depth of the input image by computing the dot product between filter weights and the input pixel values. These values are then summed with their combined value outputting a summed activation. This summed activation is subsequently transformed by applying an activation function in order to allow the subsequent use of stochastic gradient descent with backpropagation. Backpropagation computes the gradient of the loss function in order to learn and update weights and biases in order to minimize the loss function, resulting in a more accurate identification of underlying features that best represent the input. If, however, the filters have been predefined to search for specific local patterns, such as edges in the case of an edge detection filter for example, then the gradient of the loss does not need to be learnt. For CNNs, the most commonly used activation function is the Rectified Linear Unit (ReLU), which is a calculation that returns input values if they are positive (>0.0) or returns a value of zero if the input value is 0.0 or less [90,91]. The bias is a learnable value that can be thought of as a threshold applied to the activation function in order for the activation function to return values that are most relevant to extracting the underlying features of the input layer. The output of

a convolution layer is an activated feature map that characterises features detected by the filters.

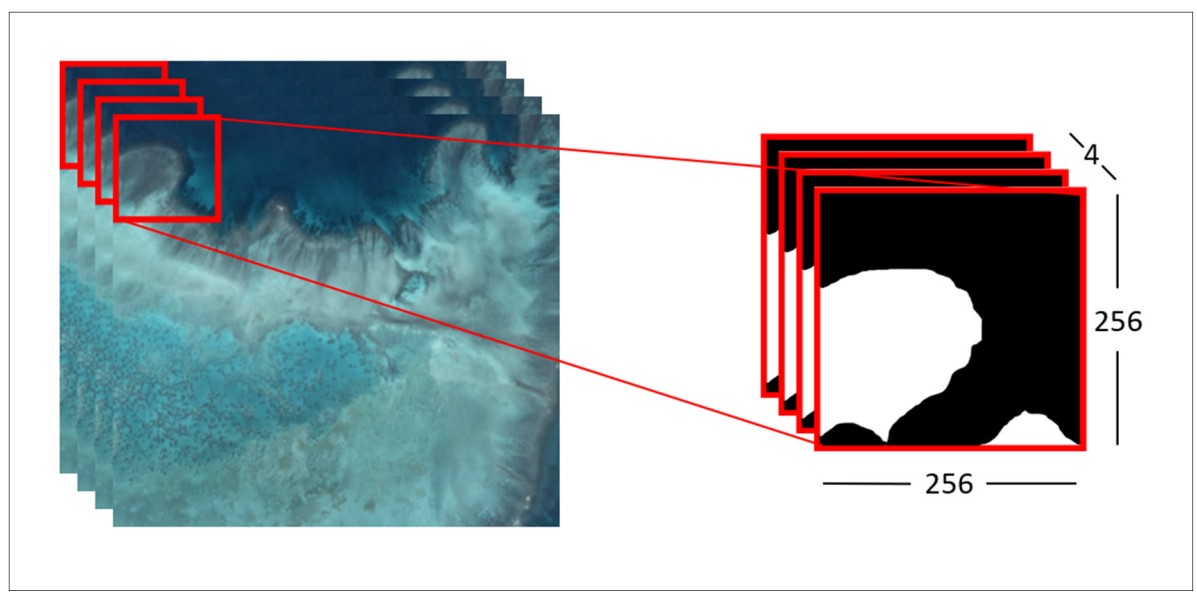

**Figure 7.** Example of a 256 × 256 × 4 training samples representing the class 'coral' (white) used as input to a CNN. The values 256 × 256 represent height and width in pixels while 4 represents the depth/number of bands in the multispectral image. The 256 × 256 × 4 sample is first extracted from the satellite image on the left. Next, all 'coral' pixels (white) are masked in order to assign the value of 0 to all non-coral pixels (black) from the training sample.

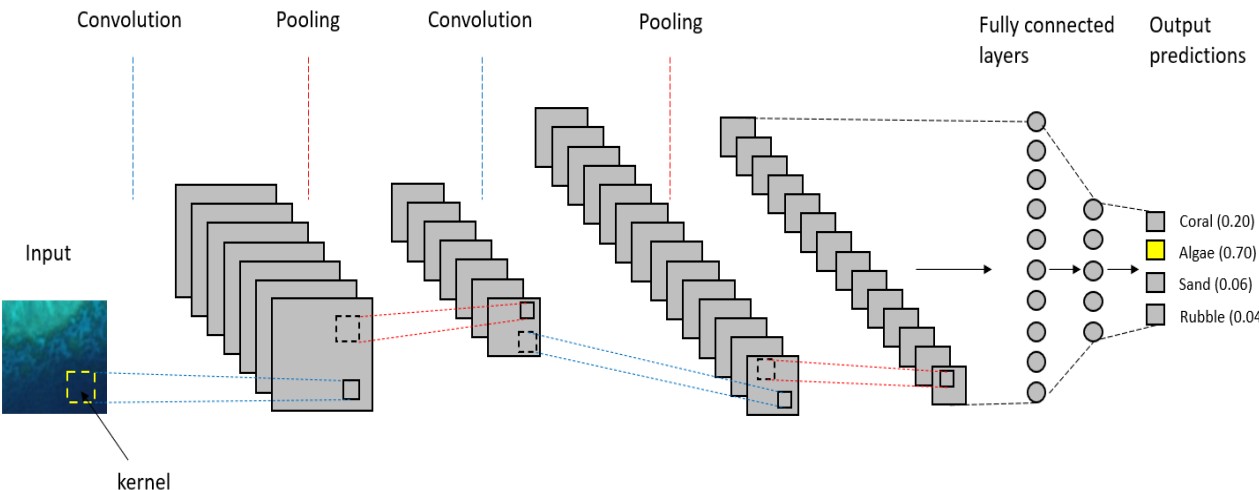

**Figure 8.** Illustration depicting a generic convolutional neural network consisting of convolutional, pooling, and fully connected layers.

In addition to the convolution operation, CNNs usually consist of pooling operations that are most commonly applied after a convolution operation. Pooling operations are used to reduce the dimensions (number of pixels) of the output feature map from the previous convolutional layer. For example, the max pooling operation uses a 2 × 2 filter applied across the height, width, and depth of the output feature map of the previous convolutional layer and outputs the maximum value of each channel. Unlike most convolution operations, max pooling uses a filter that is 2 × 2 in size rather than 3 × 3, and also uses a stride of two, therefore reducing the dimensionality of the feature map by a factor of two. Furthermore, max pooling filters do not learn weights like a convolutional filter. Max pooling, for

example, does not have weights but instead takes the maximum value from each $2 \times 2$ filter covering the region of the feature map from the previous convolutional layer as its output. The resulting feature map after max pooling, therefore, only contains the most prominent features from the previous feature map.

A key difference between MLC, *k*-NN, SVM, and RF classification algorithms compared to CNNs is the incremental, layer by layer way in which CNNs extract increasingly complex representations from inputs and the fact that these representations are learned jointly. This means that, whenever the framework updates one of its internal features, all dependent features automatically adapt to this update without manual human intervention. This therefore allows CNNs to consist of tens to even hundreds of successive layers of representations [79]. In contrast, traditional machine-learning algorithms usually only transform the input data into one or two successive layers of representations such as decision trees in the case of RFs, or high-dimensional non-linear projections in the case of SVMs [79]. CNNs can therefore extract much more complex representations of the inputs that can then be used for supervised classification at the end of the network (Figure 8).

### 2.2.1. Fully Convolutional Neural Networks

While there are CNN frameworks capable of object detection and image classification, it is image segmentation frameworks that are most suited to mapping coral reef benthic composition using remotely sensed imagery. Image segmentation frameworks assign a class label based on a probability to each individual pixel. There have been a number of different deep learning-based approaches identified for image segmentation. For example, Cireşan et al. [92] developed a deep neural network (DNN) to segment neuronal membranes in electron microscopy images based on a sliding-window approach. For this, a patch (window) is placed around a pixel and a DNN is then used to classify the central pixel within each patch. Farabet et al. [93] developed a multiscale feature extraction framework for scene labelling (labelling each individual pixel based on the class it belongs to), and Pinheiro and Collobert [94] demonstrated the use of recurrent convolutional neural networks (RCNN) for scene labelling. Another approach was developed by Long et al. [95] using a fully convolutional neural network (FCN) framework for semantic segmentation; this surpassed previous approaches in terms of accuracy and learning speed. Building upon the FCN framework of Long et al. [95], Ronneberger et al. [96] developed an FCN they called U-Net (Figure 9), which was designed to work with very few training images and yields more precise segmentations.

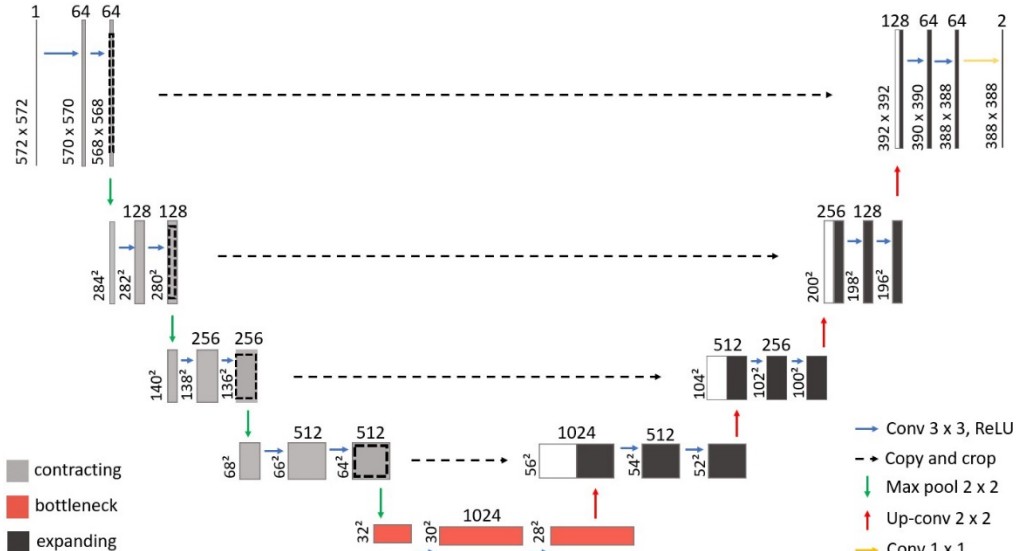

**Figure 9.** Illustration of U-Net based on the illustration of the U-Net framework by Ronneberger et al. [96].

Figure 9 illustrates the U-Net framework developed by Ronneberger et al. [96]. It starts with a contracting path that resembles a typical CNN framework, illustrated as contracting blocks in Figure 9, and consisting of convolutional and max pooling layers. Next, the contracting path transitions into a bottleneck consisting of two convolution layers followed by a ReLU and one up-convolution layer that initiates the expansive path. After each up-convolution the input is concatenated with a cropped version of the input from the corresponding block in the contracting path by using skip connections. The up-convolutions allow the framework to propagate contextual information to higher resolution layers, while the concatenation with corresponding contracting layers via skip connections allows for a more precise localisation [96]. At the final layer of the framework, the input passes through a $1 \times 1$ convolution with the number of output feature maps being equal to the number of classes that are being segmented. A pixel-wise softmax with a cross-entropy loss function is applied over the final feature map. Since its inception in 2015, U-Net has been used in a range of remote sensing publications [89,97–99].

2.2.2. Convolutional and Fully Convolutional Neural Networks Applied to Coral Reef Benthic Composition Mapping

Recently, a small number of publications have used CNN and FCN frameworks to map coral reef benthic composition using multispectral satellite imagery [31–33,80]. Mohamed et al. [33] used a simple CNN framework consisting of seven layers in order to classify seven coral reef benthic habitat classes located at Shiraho and four seagrass classes located at Fukido on Ishigaki, Japan, using pansharpened QuickBird (0.6 m) and GeoEye-1 (0.5 m) imagery, respectively. The inputs to their CNN framework were 1500 image patches that were each $2 \times 2$ pixels, which were placed around each correctly labelled in situ reference image location (an individual pixel) in horizontal and vertical directions. The results of their CNN frameworks showed an overall accuracy of 90% for mapping seven benthic classes at Shiraho and 91% for mapping four seagrass classes at Fukido.

In order to overcome the problem of spectral similarity between coral reef benthic classes, Wan and Ma [80] developed a CNN–SVM method that used spectral and multi-scale spatial information to map three different reef substrate classes (reef-depositional area, reef-clumping area, and submerged reef) and also 'seawater' using WorldView-2 (1.9 m) and Gaofen-2 (3.2 m) imagery covering Qilianyu Island, South China Sea. In addition to spectral and spatial information, Wan and Ma [80] use bidimensional empirical mode decomposition (BEMD) to extract multiscale information to incorporate into their CNN–SVM. The CNN is used to extract features of the four classes while the SVM is used in order to classify these features [80]. To determine the utility of their CNN–SVM framework, Wan and Ma [80] compared it to four other frameworks: SVM, RF, CNN, and CNN–RF, with the CNN–SVM achieving the highest overall accuracies (Table 2).

In relation to spatio-temporal generalisation, two publications have tested the utility of FCNs applied to coral reef benthic mapping [31,32]. Li et al. [32] developed an object-based fully convolutional neural network (FCN), which is similar to the U-Net framework. Unlike U-Net, however, for the contracting path Li et al. [32] use ResNet-50 adapted from [100], and for the expanding path a RefineNet [101,102] framework that uses residual convolution units (RCUs). Chained residual pooling (CRP) structures during the upsampling procedure allow additional context to be inferred. After initial segmentation using their FCN framework, they then used classified pixels in order to train a pixel-based $k$-NN algorithm based on the nearest ten points to classify coral, sediment, and seagrass. A conditional random field (CRF) on the $k$-NN algorithm was applied afterwards in order to reduce noise in the $k$-NN output and refine the boundaries.

Li et al. [32] refer to their framework as NeMO-Net and tested its capability by classifying five scenes, each with $256 \times 256$ patches covering Circa Island, Fiji, using WorldView-2 (1.9 m) imagery, and seven scenes, each with $256 \times 256$ patches using PlanetScope (3.7 m) imagery. The scenes were acquired over different dates and therefore significant spectral variations were present. NeMO-Net was compared to three other image FCN segmen-

tation frameworks: VGG16–FCN, DeepLab, and SharpMask. The results showed that NeMO-Net with the *k*-NN applied to coral, sediments, and seagrass achieved the highest overall accuracy for both WorldView-2 and PlanetScope imagery (Table 2). In order to test if NeMO-Net is capable of spatio-temporal generalisation using WorldView-2 imagery, nine different patches each $256 \times 256$ covering a different island (Fulaga, Kabara, Mago, Matuka, Moala, Nayau, Totoya, Tuvuca, and Vanua Vatu) on different days were tested and compared to the three other FCN frameworks. The results showed that all four FCN frameworks are capable of spatio-temporal generalisation based on overall accuracies >77%, with NeMO-Net achieving the highest (85%).

In order to address the problem of limited training data for coral reef benthic mapping algorithms, Asanjan et al. [31] developed a framework they refer to as LAPDANN, which can use labeled samples from high-resolution WorldView-2 imagery (1.9 m) in order to accurately predict benthic classes in medium-resolution Sentinel-2 imagery (10 m); the classified Sentinel-2 imagery can then be downscaled to match the WorldView-2 spatial resolution of 1.9 m. LAPDANN consists of an improved version of a Domain Adaptation Neural Network (DANN) adapted from Ganin et al. [103] that is capable of learning from source domains and subsequently transferring learnt information to the target domain by implementing a three-part neural network consisting of: a generative network to extract domain invariant features, a U-Net framework to segment and classify input images, and a domain discriminative network to discern features from the source and target domains [31]. Next, a Laplacian Generative Adversarial Network (LAPGAN) framework proposed by Denton et al. [104] is adapted in order to downscale the segmented Sentinel-2 imagery to higher resolution (1.9 m) coral reef maps.

The results showed an overall accuracy of 86% for ten habitat classes consisting of a mix of benthic and terrestrial classes when training and testing on Peros Banhos Island in the Indian Ocean [31]. When testing the framework's ability to generalise to new geographical regions by training LAPDANN on data from the Indian Ocean and Pacific Oceans simultaneously (information on number of samples or which reefs in the Pacific Ocean are used for training are not included), the results showed an overall accuracy of 47%. Although this lower overall accuracy indicates an inability to generalise, Asanjan et al. [31] anticipate a higher overall accuracy provided more training data capturing variations in sensing conditions over multiple islands is used. They also tested the ability of LAPDANN to generalise by training on a subset of islands and then testing on an island that is new. The overall accuracy for this experiment, however, is not reported in Asanjan et al. [31].

*2.3. Change Detection*

2.3.1. Coral Reef Benthic Change Detection Methods

Mapping changes in coral reef benthic composition using multi-temporal satellite imagery is an inherently difficult task because atmospheric, water surface, water depth, and water clarity conditions may differ between multi-temporal satellite imagery. Compared to coral reef benthic composition mapping, there are a relatively small number of publications (Table 2). This is likely due to the lack of available in situ reference data sets that have been acquired consistently over such long timeframes. Benthic change detection publications can generally be separated into either pixel-based change detection (PBCD) or object-based change detection (OBCD).

PBCD methods, in general, first map benthic composition for each image separately by using either an unsupervised or supervised pixel-based machine-learning classification algorithm [22,23,105–111], or spectral unmixing [112]. The most used pixel-based machine-learning algorithm in PBCD is the Mahalanobis distance classification algorithm, which is used by 15% of publications in Table 3, followed by MLC and SVM, each used by 12%.

OBCD approaches on the other hand usually first generate segmented objects using either manual delineation [14,24,113] or specialized object-based software such as Trimble eCongition followed by the use of a machine-learning classification algorithm such as

MLC [105] or RF [114,115]. Aside from manual polygon delineation and class assignment, which is used by 12% of publications in Table 3, the RF algorithm is the most used object-based machine-learning algorithm in OBCD, used by 8% of publications.

Once coral reef benthic composition has been mapped within each multi-temporal image, in order to subsequently identify changes, the most common approach is to use post-classification comparison change detection (PCCCD), which overlays classified images in order to identify changes between either individual pixels (post-classification pixel-based change detection (PC-PBCD)) or class objects (post-classification object-based change detection (PC-OBCD)). Alternatively, pixel-based modelling [116], simulation [117], and statistical analysis [118,119] approaches have been used to identify benthic change. These approaches, however, are usually based on expert knowledge and may even require coincident classified benthic maps to validate the models' performance [116]. This makes them prone to the similar scalability limitations of the PBCD and OBCD approaches.

The accuracies of PC-PBCD and PC-OBCD, are dependent upon the accuracy of the initial benthic composition classification; therefore, the same primary limitation present in benthic composition mapping (the inability to spatially or temporally generalise) is also present in PC-PBCD and PC-OBCD. A further limitation with PC-PBCD or PC-OBCD is the fact that image classification occurs separately between images, thus leaving the change detection output maps prone to image misregistration errors. These errors are more pronounced in PC-PBCD compared to PC-OBCD, with PC-PBCD also being prone to the 'salt-and-pepper effect' [50,115]. However, PC-OBCD accuracy is dependent on how well the objects resulting from OBIA segmentation represent the underlying benthic classes. Therefore, the choice of an optimal scale parameter used to determine the size of objects resulting from the segmentation algorithm is very important [120].

An alternative approach to PC-OBCD, which reduces image misregistration errors, is multi-temporal image object analysis (MTOA), which simultaneously segments each image within a multi-temporal data set before using a machine-learning algorithm to predict benthic change type [115]. Zhou et al. [115] developed a multi-temporal OBCD (MT-OBCD) method that simultaneously segments images before using an RF algorithm to predict change types (i.e., '*reef sediment extension*', '*algae grow*'). To identify the optimal scale parameter, Zhou et al. [115] used the Estimate Scale of Parameter Tool that is based on the rate of local variance concept (ROC-LV). In this, the scale parameter is objectively determined by automatically increasing the local variance (LV) incrementally until the ROV-LV reaches a peak. At that point, it is considered to be the optimal scale parameter [74]. They applied this MT-OBCD method to Taiping Island, Zhongye Island, and two coral reef sites on the Barque Canada Reef in the South China Sea to predict four different coral reef benthic change types using QuickBird and World View-2 satellite imagery. They achieved an overall accuracy >90% for each site.

**Table 3.** Coral reef benthic change detection publications summary.

| Authors | Pixel or Object-Based | Time-Series | Classification Method | Supervised or Unsupervised | Number of Classes Mapped | Change Detection Method |
|---------|----------------------|-------------|----------------------|---------------------------|-------------------------|------------------------|
| [121] | Pixel-based. | 2015–2016 | Radiometric normalization with pseudo invariant features (PIFs), multi-temporal depth invariant indices (DII), followed by SVM. | Supervised. | 1 (bleached coral). | PCCCD. |
| [122] | Object-based. | 2017–2019 | Unsupervised ISODATA classification. | Unsupervised. | 4 | PCCCD. |

**Table 3.** *Cont*.

| Authors | Pixel or Object-Based | Time-Series | Classification Method | Supervised or Unsupervised | Number of Classes Mapped | Change Detection Method |
|---|---|---|---|---|---|---|
| [23] | Pixel-based. | 2000–2014, 2002–2014, 2001–2015 | Pixel-based-SVM. | Supervised. | 2 | PCCCD. |
| [112] | Pixel-based. | 2009–2015 | Spectral linear unmixing using IDL CONSTRAINED_MIN optimization algorithm followed by assigning class thresholds. | Supervised. | 13 | PCCCD. |
| [113] | Object-based. | 2001–2015 | Manual polygon delineation. | Supervised. | 8 (habitat scenario trajectories). | PCCCD. |
| [114] | Object-based | 2014–2016 | Unsupervised IRMAD to detect areas of change, OBIA–RF to classify classes, overlaying images to perform supervised change detection. | Unsupervised and Supervised. | 10 habitat classes and 5 classes of change type. | PCCCD. |
| [115] | Object-based (multiresolution segmentation) and pixel-based. | 2013–2015 | OBIA–RF change prediction, pixel-based-RF change prediction. | Supervised. | 5 change types (i.e., reef sediments extension). | MT-OBCD. |
| [123] | Pixel-based. | 1994–2014 | Unsupervised Iterative self-organizing class analysis (ISOCLASS) followed by supervised reclassification based on visual interpretation. | Unsupervised and Supervised. | 5 | PCCCD. |
| [110] | Pixel-based. | 2001–2014 | SVM | Supervised. | 11 | PCCCD. |
| [22] | Pixel-based. | 1987–2013 | ISODATA clustering followed by unsupervised $k$-means classification; MLC. | Unsupervised and Supervised. | 10 unsupervised, 5 supervised. | PCCCD. |
| [124] | Pixel-based. | 2005–2008 | MLC for mapping 5 classes then '*differences in reflectance values between two images within the coral classes were used to detect bleached corals*.' | Supervised. | 5 | PCCCD. |
| [24] | Object-based (manually delineated polygons). | 1972–2007 | Photo-interpretation based on manual polygon delineation. | Supervised. | 3, 19, and 42 (based on level 1, 2, and 3 maps). | PCCCD. |

**Table 3.** *Cont.*

| Authors | Pixel or Object-Based | Time-Series | Classification Method | Supervised or Unsupervised | Number of Classes Mapped | Change Detection Method |
|---|---|---|---|---|---|---|
| [105] | Pixel-based and object-based. | 2002–2004 | Post-cyclone coral community structure maps: Photo-interpretation based on manual polygon delineation, pixel-based MLC, OBIA-MLC; Pre-cyclone community maps: *post-cyclone coral community structure classes were used to label pre-cyclone polygons based on consistent colour and texture visible on the images, and also accounting for proximity* [105]. | Supervised. | 20 | PCCCD. |
| [111] | Pixel-based. | 1991–2002 | Parallelepiped classification. | Supervised. | 6 | PCCCD. |
| [14] | Object-based (manually delineated polygons). | 1973–2007 | Photo-interpretation based on manual polygon delineation. | Supervised. | 15 | PCCCD. |
| [106] | Pixel-based. | 1984–2002 | Mahalanobis distance classification. | Supervised. | 4 | PCCCD. |
| [116] | Object-based (timed automata model), Pixel-based (minimum distance classification). | 2002–2004 | A combined generic timed automata model of reef habitat trajectories and classified remotely sensed imagery based on MD classification. | Supervised. | 36 (habitat classes). | PCCCD, Modelling (generic timed automata). |
| [119] | Pixel-based. | 1990–2001 | Unsupervised ISODATA classification followed by calculating the median coefficient of variation (COV). Images were then segmented by habitat to create habitat masks and also segmented by representative quadrants. The median COV for each habitat and quadrat were calculated before performing a Kruskall–Wallis nonparametric test to determine whether differences between the median COV values were significant at the 0.05 level. | Unsupervised and Supervised. | 6 class habitat map, test for significant differences. | PCCCD, statistical analysis. |

**Table 3.** *Cont.*

| Authors | Pixel or Object-Based | Time-Series | Classification Method | Supervised or Unsupervised | Number of Classes Mapped | Change Detection Method |
|---|---|---|---|---|---|---|
| [125] | Pixel-based. | 1987–2000 | Multi-component change detection: image differencing to determine areas of significant change followed by MLC. Images were then '*combined*' to create a 'from-to change map.' | Supervised. | 4 benthic classes each with 6 possible change types. | PCCCD. |
| [126] | Pixel-based. | 1991–2003, 2000–2001 | Unsupervised K-means clustering followed by PCA. | Unsupervised. | 3 | PCCCD. |
| [117] | Pixel-based. | 1984–2000 | Radiative transfer simulation and also an image normalisation method [127] followed by digital number comparison. | Supervised. | 2 (radiative transfer simulation: bleached coral, healthy coral), 2 (normalisation method: slightly or non-bleached, severely bleached). | PCCCD (normalisation method), Modelling (radiative transfer simulation). |
| [109] | Pixel-based. | 1984–2000 | Mahalanobis Distance classifier. | Supervised. | 4 | PCCCD. |
| [108] | Pixel-based. | 1981–2000 | Mahalanobis Distance classifier. | Supervised. | 4 | PCCCD. |
| [128] | Pixel-based. | 1998 (February)–1998 (August) | Image differencing based on mean (3 × 3) filtering, PCA, difference between local variation calculated as a standard deviation in a 3 × 3 neighbourhood. | Supervised. | 1 (bleaching detection). | PCCCD. |
| [118] | Pixel-based. | 1994–1996 | Getis Statistic. | Supervised. | Test for significant difference. | PCCCD—Spatial autocorrelation. |
| [107] | Pixel-based. | 1984–1999 | Mahalanobis Distance classification. | Supervised. | 4 | PCCCD. |

(Unsupervised ISODATA Clustering [22,122] and photointerpretation based on manually delineating polygons [14,24] has been grouped into 'object-based' even though these polygons are not derived by utilizing a segmentation algorithm such as those derived from OBIA. Accuracies are not reported since no consistent accuracy metric (i.e., overall accuracy) is used between all publications. Sensor/spatial resolution of imagery used is also not reported as it is in Tables 1 and 2 since it is not directly relevant in terms of identifying the change detection methodology and since accuracies are not reported (i.e., sensor/spatial resolution affects accuracies, therefore, should be included with accuracies). Pixel-based modelling [116], simulation [117] and statistical analysis [118,119] approaches have been grouped into PCCCD).

Due to the logistical complexities inherent to collecting in situ reference data from coral reef sites, there are very few consistent long-term in situ reference data sets available. Therefore, to map benthic changes over long time periods (>30 years), scientists have had to collate any available in situ reference data and imagery. This may not be consistent in terms of in situ reference data acquisition methodology, imagery type (i.e., aerial photographs

and satellite imagery), image extent, and imagery acquisition dates. For example, in order to map fifty years of benthic habitat changes occurring on the outer reef flats of Grand Recif of Toliara in southwest Madagascar, Andréfouët et al. [129] used an imagery data set consisting of a combination of irregularly dated historical aerial photographs and satellite imagery from different satellite sensors (Quickbird, WorldView-2, Landsat-5, Landsat-7). In addition, there was a training and validation data set consisting of a combination of in situ reference data derived from benthic photo-transects as well as data derived from reference to hand drawn maps. Such mixed data sets make it difficult to train machine-learning algorithms to produce accurate benthic maps that can be used in PCCCD. In such instances, photointerpretation based on manual polygon delineation has been demonstrated as being a suitable alternative [14,129], although as previously mentioned, it is difficult to replicate since it is relatively subjective.

Because of the increase in high temporal and high spatial resolution satellite imagery now available to coral reef remote sensing scientists, it is now possible to ensure the consistent spatial and spectral resolution of imagery to be used for coral reef benthic change detection. For example, Planet Labs provide high spatial resolution (<5 m) multispectral satellite imagery covering the entire Earth's landmass with a daily temporal resolution. However, even with high temporal resolution satellite imagery, for a benthic change detection methodology to be relatively objective in nature compared to manual delineation, coincident in situ reference data are required. This requirement remains a primary limitation in relation to the spatio-temporal scalability of coral reef benthic change detection regardless of whether PC-PBCD, PC-OBCD, or MT-OBCD is used.

To address this limitation, Gapper et al. [23] investigated the use of an SVM algorithm applied to binary (coral and not coral) PC-PBCD and tested its spatial generalisability when applied to new reef sites that contained no site-specific in situ reference data. Four reef sites were used with an iterative classification process applied whereby each reef was first classified individually (training and test data derived from the same reef) using a site-specific SVM algorithm, then subsequently applying an SVM algorithm that has been trained on data derived from the three other reefs that they refer to as a controlled parameter cross-validation (CPCV) procedure. Site-specific overall accuracies ranged from 69% to 88% while the CPCV overall accuracies ranged from 65% to 81%. Although the CPCV overall accuracies were on average 10% lower than the overall accuracies of the site-specific SVM algorithms, and the fact this approach was tested on medium resolution (30 m) Landsat-7 and Landsat-8 imagery, which only allowed for binary (coral or not coral) PC-PBCD, Gapper et al. [23] has demonstrated the ability of an SVM to spatially generalise to new reefs, while maintaining moderate to high overall accuracies.

### 2.3.2. Recurrent Neural Networks Applied to Land Cover Change Detection Using Multispectral Satellite Imagery

Deep learning frameworks are also emerging as new algorithmic approaches for change detection using multispectral satellite imagery. So far though, they have not been applied to coral reef benthic change detection. When it comes to processing multi-temporal imagery, there is a subset of deep learning frameworks known as recurrent neural networks (RNNs) that are suited to dealing with sequential time series data [34,35]. A key difference between RNN frameworks compared to the PC-PBCD, PC-OBCD, and MT-OBCD methods currently used for coral reef benthic change detection is their ability to learn information relating to the difference between corresponding pixels within multi-temporal imagery (i.e., the difference between a pixel in the time-two image compared to the corresponding pixel in the time-one image) and store that learned information as a hidden state. This hidden state allows the RNN framework to detect changes in any new corresponding pixels added to the sequence (i.e., the corresponding pixel at time-three compared to the same pixel at time-two) without further training, since it still contains the stored memory of prior inputs.

While simple RNNs can suffer from the vanishing and exploding gradient problems, recurrent neural networks composed of Long Short-Term Memory units (LSTM), first introduced by Hochreiter and Schmidhuber [130], are less prone to this problem. Each LSTM unit contains a core memory cell that is regulated by three gates: an input gate, an output gate, and a forget gate (Figure 10). The core memory cell remembers information between multi-temporal data, while the input gate controls what new information is added into the cell, the forget gate controls what information is kept within the core memory cell or forgotten, and the output gate controls the information used to compute the output activation. For multi-temporal satellite imagery, recurrent connections between LSTM units at each time step allow the model to learn information relating to the difference between corresponding pixels (pixel from the time-two image and the corresponding pixel from the time-one image) that can subsequently be used for binary and/or multi-class change detection [34].

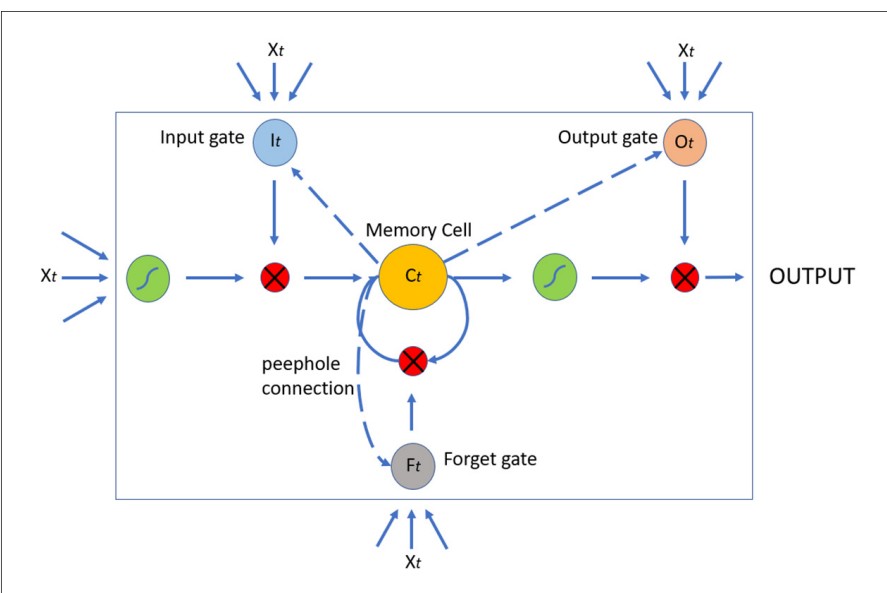

**Figure 10.** Basic conceptualisation of an LSTM unit where Xt represents the input vector at time t. It, Ot, Ft, and Ct represent the input gate, output gate, forget gate, and core memory cell, respectively. Blue arrows indicate directional information flow. Dotted blue arrows indicate peephole connections. This illustration is based on the illustration of an LSTM unit in Figure 5 depicted in Lyu et al. [34].

Lyu et al. [34], who were perhaps the first to use an RNN framework for the purpose of land cover change detection using multispectral satellite imagery, detected binary (changed and unchanged) and multi-class (i.e., city expansion, changed soil region, and changed water areas) changes at three different city sites in China using an LSTM framework they refer to as REFEREE (Figure 11). Looking at their REFEREE framework in Figure 11, it can be seen that first the input layer receives a 6-band pixel (pixel from a multispectral image with six spectral bands) that has been extracted from the T1 image (image at time-1). Next, the hidden layer receives this input and calculates its state information (which can be thought of as information about this particular pixel at this particular point in time) and stores this information. It is important to note that the hidden layer consists of LSTM units. The corresponding 6-band pixel from the T2 image (image at time-2) is then input to the hidden layer simultaneously, whereby change information between these two corresponding pixels can be learned by the current hidden layer. The label of '*changes*' or '*no changes*' can then be predicted in the decision layer.

Lyu et al. [34] compared their LSTM framework to four conventional change detection methods (change vector analysis (CVA), principal component analysis (PCA), iteratively reweighted multivariate alteration detection (IRMAD), and Slow Feature Analysis (SSFA)) for their binary change detection and three methods for their multi-class change detection



(support vector machine (SVM), Decision Tree (DT), and a CNN). The results of their binary change detection experiments showed that their LSTM framework achieved the highest overall accuracies at three sites (the overall accuracies were all 98%), outperforming CVA (70–87%), PCA (74–90%), IRMAD (84–94%), and supervised slow feature analysis (SSFA) (95–98% (When overall accuracies are rounded to the nearest whole number of the LSTM of Lyu et al. [34], the highest is 98%, which is the same as the highest overall accuracy for SSFA, however, when rounded to the nearest one decimal place the LSTM is higher (98.4%) compared to the SSFA (97.6%)). Their multi-class change detection experiments resulted in their LSTM framework again achieving the highest overall accuracies (95–96%), outperforming CNN (92–93%), SVM (80–84%), and DT (70–71%). Lyu et al. [34] also demonstrated that their LSTM framework is capable of spatio-temporal generalisation when applied to binary change detection. This means it can transfer a learned change rule (information relating to the difference between corresponding pixels from multi-temporal imagery) to new multi-temporal imagery of a different city that the LSTM framework has not been trained on without requiring any extra learning processes. All imagery used in their transfer experiments had a similar spectral resolution (six bands) and the same spatial resolution (30 m). Their transfer experiments for binary change detection using different numbers of training samples ranging from 200–1000 resulted in overall accuracies ranging from 72–97%.

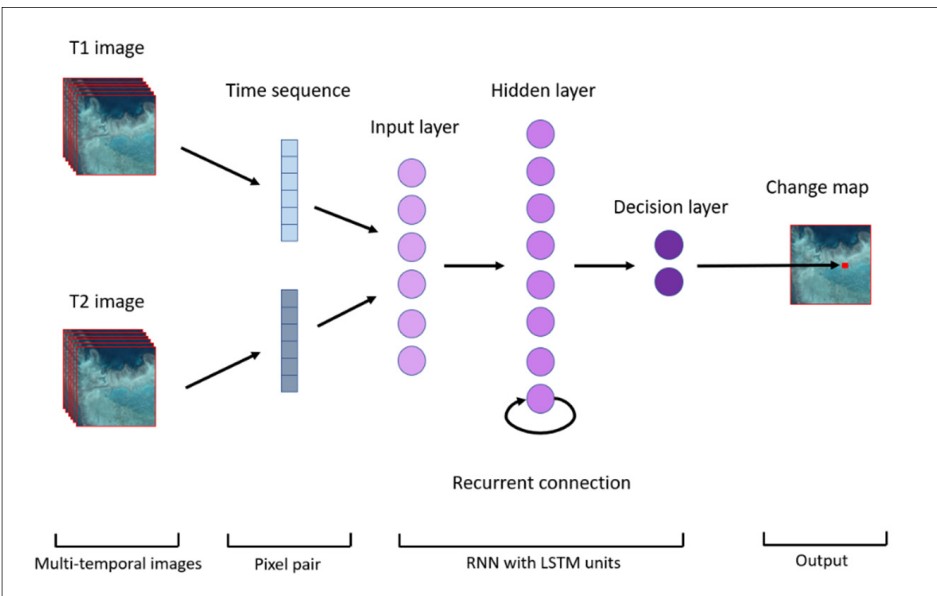

**Figure 11.** Illustration of the RNN-LSTM framework developed by Lyu et al. [34], which they refer to as REFEREE. The hidden layer consists of LSTM units as illustrated in Figure 10. This illustration is based on the illustration in Figure 4 depicted in Lyu et al. [34].

Another binary and multi-class land cover change detection framework was developed by Mou et al. [35] who used a recurrent convolutional neural network (ReCNN) that extracts joint spectral-spatial-temporal features from bi-temporal multispectral images. Their ReCNN combined the ability of a CNN to extract contextual features with an RNN's ability to model the temporal correlation between multi-temporal data. Their publication compared their ReCNNs' overall accuracy with the overall accuracy of six conventional land cover change detection methods (CVA, PCA, multivariate alteration detection (MAD), IRMAD, DT, and SVM). In addition, the LSTM framework referred to as REFEREE that was developed by Lyu et al. [34] was also compared, as well as three variations in the recurrent sub-network of their ReCNN framework. One had a fully connected RNN (ReCNN–FC), a second had LSTM units as the recurrent component (ReCNN–LSTM), and a third had Gated Recurrent Units (ReCNN–GRU). For both binary and multi-class change detection

experiments, the overall accuracy of their ReCNN–LSTM achieved the highest overall accuracies ranging from 98–99%, which proved superior to the six conventional change detection methods (75–96%), as well as the REFEREE model of Lyu et al. [34] and the three variations in the recurrent sub-network of their ReCNN (95–99% (When the ReCNN–LSTM and ReCNN–GRU frameworks of Mou et al. [35] are rounded to the nearest whole number of 99%, they are the same. However, in Mou et al. [35] they round overall accuracies to the nearest two decimal points which shows the ReCNN–LSTM achieving the highest (98.70) compared to the ReCNN–GRU (98.64)). However, the spatio-temporal generalisability of their ReCNN framework was not tested.

## 3. Conclusions

### 3.1. Coral Reef Benthic Mapping

In relation to mapping coral reef benthic composition using multispectral satellite imagery, Tables 1 and 2 show a total of 47 papers; 17 are pixel-based and 30 are object-based. The most used pixel-based machine-learning classification algorithm is MLC, which is used by 47% of publications in Table 1, followed by RF, SVM, and MDM in second with each being used by 12%. In relation to object-based publications, aside from OBIA followed by expert driven membership rulesets (23%), SVM is the most used machine-learning classification algorithm, being used by 17%, followed by RF and *k*-NN in second place, each being used by 12%. Based on these two tables, we can conclude that object-based machine-learning classification algorithms are more commonly used compared to pixel-based. Furthermore, object-based machine-learning algorithms are not as prone to pixel-based issues mentioned in Section 2.1.

It is not possible to directly compare all machine-learning algorithms and change the detection methods covered in this review because of inconsistencies in classification schemes. These inconsistencies can be caused by a variety of differences in imagery (spectral and spatial resolutions), in situ reference data collection methodologies, the number and types of benthic classes mapped, and accuracy assessment protocols. It is clear that currently the main limitation in relation to spatio-temporal scalability is the requirement of coincident in situ reference data. It should be noted that this review focused on overall accuracy as the metric used to determine the accuracy of each machine-learning algorithm reviewed. However, the overall accuracy does not provide insight into class-specific accuracies. Therefore, further investigation is required to determine the utility of each algorithm in relation to class-specific accuracies.

In relation to spatio-temporal generalisation, only three publications in Tables 1 and 2 tested the generalisability of machine-learning algorithms [31,32,41], and one is still yet to verify the accuracy of their OBIA–RF s generalisability [59]. Based on the findings of these four publications, we believe there are two current approaches that have the potential to increase the spatio-temporal scalability of coral reef benthic mapping. The first is to use expert-derived training and validation samples [59,131]. This approach is necessary because existing pixel- and object-based machine-learning classification algorithms require coincident in situ reference data. Therefore, to increase their spatial scalability to a global scale, in situ reference data sets need to increase to a global scale as well. While [59], as part of the Allen Coral Atlas, have adopted this approach developed by Roelfsema et al. [131] on a global scale, they do not, however, report the accuracy of their object-based RF algorithm in areas outside the extent of their reference data; therefore, it remains currently unvalidated in those areas.

The second approach is to identify machine-learning algorithms capable of spatio-temporal generalisation. To this end, Gapper et al. [41] used pixel-based Linear Discriminant Analysis to map two benthic classes (coral and algae/sand). However, this was based on low resolution (30 m) Landsat-8 imagery, making it prone to pixel-based issues and was limited in terms of the number of benthic classes that could be mapped. Li et al. [32] demonstrated that na FCN framework combined with a *k*-NN classification algorithm can achieve high overall accuracies in relation to mapping three benthic classes (coral, seagrass,

and sediment) using high resolution WorldView-2 (1.9 m) and PlanetScope (3.7 m) imagery and also demonstrated spatio-temporal generalisation. Furthermore, Asanjan et al. [31] developed the LAPDANN framework that can use labeled samples from high resolution WorldView-2 imagery (1.9 m) to accurately predict five benthic classes (reef crest—coralline algae ridge, fore-reef, back-reef pavement or sediment, back-reef coral framework, and seagrass meadows) in medium resolution Sentinel-2 imagery (10 m) with a high overall accuracy. This demonstrates an ability to generalise to different sensors and spatial resolutions. Based on these papers, the deep learning frameworks of Li et al. [32] and Asanjan et al. [31] appear to hold the most potential in relation to increasing the spatio-temporal scalability of coral reef benthic mapping.

### 3.1.1. Coral Reef Benthic Change Detection

In relation to change detection, Table 3 shows 26 papers that have mapped coral reef benthic change over time. Most (54%) of these use some form of pixel-based PCCCD. The most used pixel-based machine-learning algorithm used in PBCD is the Mahalanobis distance classification algorithm, used by 15% of publications in Table 3 followed by MLC and SVM, each used by 12%. Aside from manual polygon delineation and class assignment which is used by 12% of publications in Table 3, the RF algorithm is the most used object-based machine-learning algorithm in OBCD, used by 8% of publications. None of the 26 publications in Table 3 have demonstrated spatio-temporal generalisation. However, in this review we did identify the LSTM framework of Lyu et al. [34] that has proven superior in terms of overall accuracy compared to post-classification land-cover change detection methodologies. It has also demonstrated spatio-temporal generalisability. Coral reef benthic mapping and change detection may be a more difficult task to accomplish compared to land cover mapping and change detection because of the further complication of the water surface and water column light scattering and absorption affecting spectral reflectance. However, we believe it is reasonable to hypothesize the potential utility of the LSTM framework for binary and/or multi-class coral reef benthic change detection using multispectral satellite imagery.

### 3.1.2. Future Research

Given the increase in the number of publications using deep learning frameworks for the purpose of mapping and change detection using remotely sensed imagery, we acknowledge that other deep learning frameworks may also be candidates for spatio-temporal generalisation in relation to coral reef benthic mapping and change detection. However, based on examples in this review we believe there are four potential areas for further investigation relevant to increasing the spatio-temporal scalability of coral reef benthic mapping and change detection. These potential areas are to determine: (1) whether the FCN framework of Li et al. [32] (NeMO-Net) can generalise to different biogeographical regions that might exhibit different benthic compositions and spatial complexity; (2) whether the LAPDANN framework of Asanjan et al. [31] can spatially generalise when more training data-capturing variations in sensing conditions over multiple reefs is used; (3) whether the OBIA–RF of Lyons et al. [59], using the expert-derived training and validation data approach of Roelfsema et al. [131], can generalise to areas outside the extent of their reference data; and (4) the utility of the LSTM framework of Lyu et al. [34] for coral reef benthic change detection.

**Author Contributions:** Conceptualization, C.B., B.B. and A.N.; methodology, C.B., A.N. and B.B.; formal analysis, C.B.; data curation, C.B.; writing—original draft preparation, C.B.; writing—review and editing, C.B., A.N. and B.B. All authors have read and agreed to the published version of the manuscript.

**Funding:** This research received no external funding.

**Acknowledgments:** The authors would like to thank Akbar Ghobakhlou, Lennard Gillman, and Ashray Doshi for their advice related to this work. The authors would also like to thank the support of the AUT Doctoral Scholarship (Fees)—School of Engineering, Computer & Mathematical Sciences.

**Conflicts of Interest:** The authors declare no conflict of interest.

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
