# Peer review of "Machine-Learning for Mapping and Monitoring Shallow Coral Reef Habitats"

_remotesensing, doi:10.3390/rs14112666_

Round 1

Reviewer 1 Report

The paper aims to illustrate the techniques and methodologies used for mapping the seabed, in particular coral reef environments.

The article illustrates the used techniques, describes the obtained results and the statistical quality of these.

While not bringing any news in terms of image processing methodologies, the work has the advantage of enclosing in a collection the references of what we can be defined as the “state of the art” on the task.

The authors conclude that some of the used methods, properly investigated, could lead to the algorithms improvement in relation to the space-time scalability of coral reef mapping and change detection, overcoming the real obstacle that remote sensing techniques applied to the sea still show today. 

Minor revision
152-153 maybe you want to say “meaning the spatial resolution of the pixel is lower than or similar to….”

Author Response

Thank you very much for reviewing my literary review. 

I do mean “higher than” in the following sentence: “First, each individual pixel needs to be represented by only one benthic class, meaning the spatial resolution of the pixel is higher than or similar to the target object.”

I’m using "higher than" to mean smaller pixel size. For example, 1-metre spatial resolution is higher than 5-metre. If the spatial resolution of the pixel is 5-metres and the target object is only 1-metre in size then there may be additional classes within the pixel.

Reviewer 2 Report

The paper provides a review on the methods for classification of coral reef benthic classes and detection of benthic changes over time using multispectral satellite imagery. The review is a valuable source for experts examining the coral reef habitats using remote sensing. Altogether 73 papers were found which were studying any of the above two problems. Tables are presented to summarise and compare the main properties of the algorithms published in the selected papers. The most relevant methods are explained in detail. Several figures facilitate understanding the algorithms. The authors examine how to improve the spatio-temporal scalability of coral reef benthic mapping in Section conclusion and select the most promising methods. 

Author Response

Thank you very much for reviewing my literary review.

Reviewer 3 Report

This is an interesting review and well written. I think it deserves to be published. 

- A quick spell checking through out the manuscript. 

- Table 2, 3: I would suggest making these table into landscape or much smaller font to fit in one page. At the moment is slightly disrupting the flow of the reading. 

Author Response

Thank you very much for reviewing my literary review. Based on your comments I’ve done a spell check throughout and made the font size smaller in Tables 1, 2 and 3.